# Usability, occupational performance and satisfaction evaluation of a smart environment controlled by infrared oculography by people with severe motor disabilities

Mariana Midori Sime[1]*, Alexandre Luís Cardoso Bissoli[2‡], Daniel Lavino-Júnior[3‡], Teodiano Freire Bastos-Filho[4]

1 Occupational Therapy Department, Federal University of Espirito Santo (UFES), Vitoria, Brazil, 2 Postgraduate Program in Electrical Engineering, Electrical Engineering Department, Federal University of Espirito Santo (UFES), Vitoria, Brazil, 3 Electrical Engineering Department, Federal University of Espirito Santo (UFES), Vitoria, Brazil, 4 Postgraduate Program in Electrical Engineering, Electrical Engineering Department and Postgraduate Program in Biotechnology, Federal University of Espirito Santo (UFES), Vitoria, Brazil

☯ These authors contributed equally to this work.
‡ ALCB and DLJ also contributed equally to this work.
* mariana.sime@ufes.br

## Abstract

A smart environment is an assistive technology space that can enable people with motor disabilities to control their equipment (TV, radio, fan, etc.) through a human-machine interface activated by different inputs. However, assistive technology resources are not always considered useful, reaching quite high abandonment rate. This study aims to evaluate the effectiveness of a smart environment controlled through infrared oculography by people with severe motor disabilities. The study sample was composed of six individuals with motor disabilities. Initially, sociodemographic data forms, the Functional Independence Measure (FIM™), and the Canadian Occupational Performance Measure (COPM) were applied. The participants used the system in their domestic environment for a week. Afterwards, they were reevaluated with regards to occupational performance (COPM), satisfaction with the use of the assistive technology resource (QUEST 2.0), psychosocial impact (PIADS) and usability of the system (SUS), as well as through semi-structured interviews for suggestions or complaints. The most common demand from the participants of this research was 'control of the TV'. Two participants did not use the system. All participants who used the system (four) presented positive results in all assessment protocols, evidencing greater independence in the control of the smart environment equipment. In addition, they evaluated the system as useful and with good usability. Non-acceptance of disability and lack of social support may have influenced the results.

**Data Availability Statement:** All relevant data are within the paper and its Supporting Information files.

**Funding:** AB and TBF received an award in Google Research Awards for Latin America, of Google Inc. The funders had no role in study design, data collection and analysis, decision to publish, or preparation of the manuscript.

**Competing interests:** Google Inc. only supported the research through Google's Latin America Research Awards (https://research.google/philosophy/) to two authors, Teodiano Freire Bastos-Filho and Alexandre Luís Cardoso Bissoli, and this does not alter our adherence to Plos ONE policies on sharing data and materials.

## Introduction

Assistive technology (AT) can be defined as an area of knowledge that includes products, resources, methodologies, strategies and services [1], or items, products and equipment acquired, adapted or modified [2], always with the aim of improving the functional performance, independence, and quality of life (QoL) of people with disabilities [1, 2].

The literature indicates that individuals with diseases or injuries that affect the central nervous system, such as Multiple Sclerosis, Amyotrophic Lateral Sclerosis, Stroke, and Cranioencephalic or Spinal Cord Injury, can present sensory, motor, language and behavioral impairments at different levels, which lead to deficits in their occupational performance for carrying out Activities of Daily Living (ADL) independently, or interacting with people and objects [3–11], making them quite dependent on family members and/or caregivers [12].

According to the International Classification of Functioning, Disability and Health (ICF), people's impairments are configured as their environments/contexts limit their activities and restrict their social participation, not favoring their functionality [13–15].

The elements that constitute the ICF's model are Health Condition, Body Functions and Structures, and Activity, Participation, and Contextual Factors (Environmental and Personal), with AT devices and resources included in Environmental Factors [13–15], which improve the functionality of people with motor disabilities and/or older people, in different areas and health conditions [4, 16–23].

However, although AT plays an important role in the recovery or improvement of the functionality of people with disabilities, the rates of abandonment and/or non-use of AT devices are high [24–28] for many reasons [19, 23, 24, 28].

Conceptual models assist researchers and professionals with making better indications and implementation of AT devices. For instance, the Human Activity Assistive Technology (HAAT) model proposes to understand the role played by AT in the lives of people with disabilities. The HAAT model is based on four elements: the Human, the Activity, the Assistive Technology, and the Context in which the other three elements are inserted. It briefly describes "someone (human) doing something (activity) in a context using assistive technology" [29] (p.7).

Thus, during the process of preparing and/or indicating an AT resource or device, it is important to understand the activity that the person wants and needs to perform, the capacities they have, and the different aspects of the context that will influence their acquisition and use. Several studies have highlighted the importance of patient/user participation in the development of AT resources or devices [22, 30–32], or in the process of defining and choosing the device that best suits their needs and of training and updating the team to evaluate and monitor the AT use [22, 24, 27, 33].

Although there are several definitions of Smart Environment (SE) [34–36], it can be defined as a space (room, house, etc.) where services (temperature, lighting, entertainment, security, etc.) and/or equipment (lamps, home appliances, alarms, etc.) are managed intelligently using technology (personal computer, tablet, smartphone, remote control, etc.), through a Human-Machine Interface (HMI), aiming to assist users or residents with their ADL and provide them with better QoL [37, 38].

Many studies have focused on the development of SEs that aim to provide greater independence for people with motor disabilities, combining their residual skills with the physical environment, since this group experiences several limitations in the use of environments and equipment control [37, 39–45]. The secondary objective is to reduce their need for assistance from caregivers or family members [45].

Despite the gradual increase in the number of these studies, only few of them have addressed the benefits of SEs for people with disabilities regarding the exercise of autonomy, i.e., freedom of opinion, choice and decision [46], improvement of performance, and usability.

The reviews by Martin et al. [47] and Brandt et al. [48] found no evidence about the effectiveness of SEs for people with disabilities. Differences in sample size, interventions, and instruments used hinder comparison between these studies, but it was possible to notice a tendency to facilitate independence, instrumental ADL, socialization, and QoL.

Marikyan, Papagiannidis and Alamanos [49] consider that there is increased research addressing SEs; however, they are restricted to three themes: they ignore the multidimensionality of the concept, disregarding the various implications, services, and user segments; focus on the functioning of technological devices, architecture and infrastructure; and are little dedicated to the perspective of users.

For the control of electronic equipment in an SE by people with motor disabilities, different ways of capturing their inputs can be used. Among them, Infrared Oculography (IROG) is a technique that has been significantly studied in computer science [42, 45, 50–52].

In IROG, a device performs eye movement tracking, calculating the point on the computer screen the user is looking at. Eye tracking devices have a video camera equipped with high resolution infrared (IR) light-emitting diodes (LED) that reflect and increase the contrast between the pupil and the iris, allowing precise pupil location and facilitating the tracking of eye movement. This movement then functions as an HMI modality, enabling users to control several applications [53–55].

This technique has proved to be one of the most indicated and useful for people with severe motor disabilities, enabling them to use HMIs in an easier, comfortable and intuitive way [56], without the need to place electrodes or equipment on their bodies. Another contributing factor is that eye movement is one of the few abilities maintained in people with severe motor disabilities [57].

Since the literature points to a lack of studies that address the effectiveness provided by AT [58, 59] as well as the importance of good assessment using valid, reliable and viable instruments, and covering various resource aspects [60], it is important that further studies addressing the effectiveness of SEs in the everyday life of people with severe motor disabilities be conducted.

The SE system used in this study was developed at the Assistive Technology Center of the Federal University of Espirito Santo (UFES), Brazil. It consists of a smart global box (gBox) coupled to a computer software that enables the user to control the TV, radio, fan and/or lighting using eye-tracking technology [50].

In this sense, the main objective of this study was to evaluate the effectiveness of the developed SE controlled through IROG for specific use by people with severe motor disabilities.

## Materials and methods

This study was approved by the Human Research Ethics Committee of Federal University of Espirito Santo, Brazil, under protocol no. 976.828, CAAE 39410614.6.0000.5060. All participants or their legal guardians signed and received a copy of the Free and Informed Consent Form, allowing the publication of their data collected in the research, as long as the confidentiality of personal information is guaranteed.

### Materials

The following materials were used in this study:

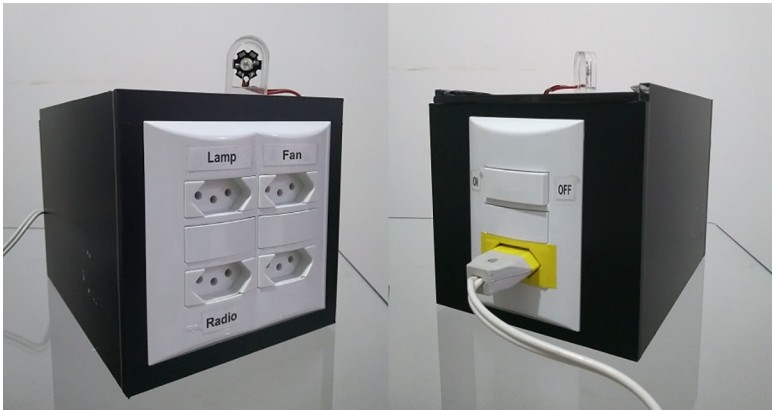

**Fig 1. gBox: Electronic module to control home devices in the SE.**

1. Notebook computer with Intel® Core™ i3-5005U processor, Windows® 10 Home Edition operating system, 4GB RAM, 500 GB HD memory, and 14" LED screen. 2. Tobii Eye Tracker 4C [61], which allows: a) booting with the computer, b) controlling with only one or both eyes, c) making movements with the head, maintaining the calibration. 3. Gaze Point software [62]: to control the mouse cursor using Tobii Eye Tracker 4C. 4. Global Box (gBox) (Fig 1): an SE controller module [50] that receives commands from the computer, via Wi-Fi, to activate home devices. 5. SE Control Interface (CI) (Fig 2) [50]: configured in a Web

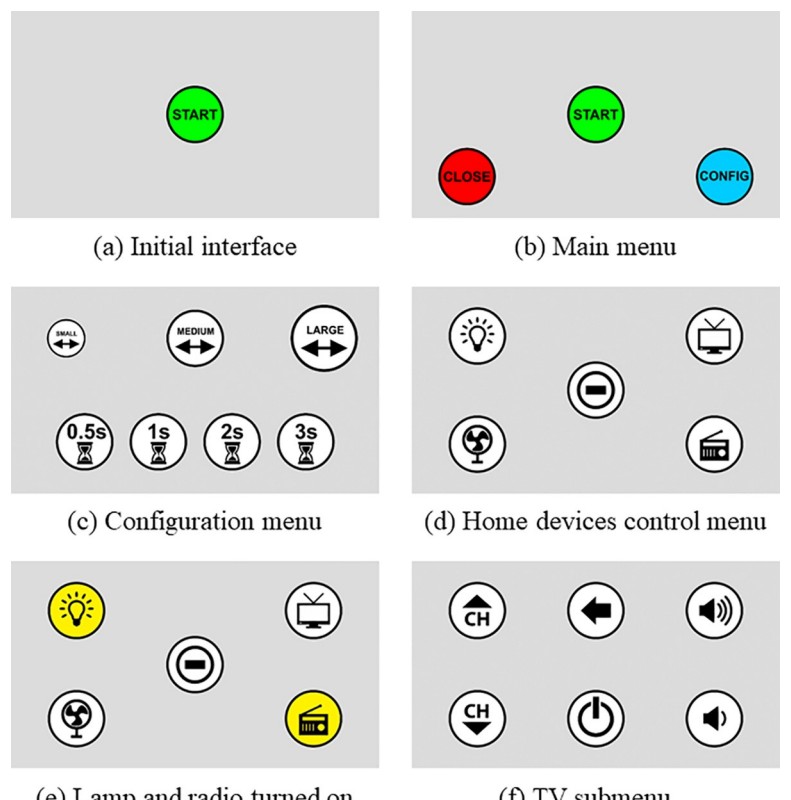

**Fig 2. User CI.** Reproduced with permission from [50].

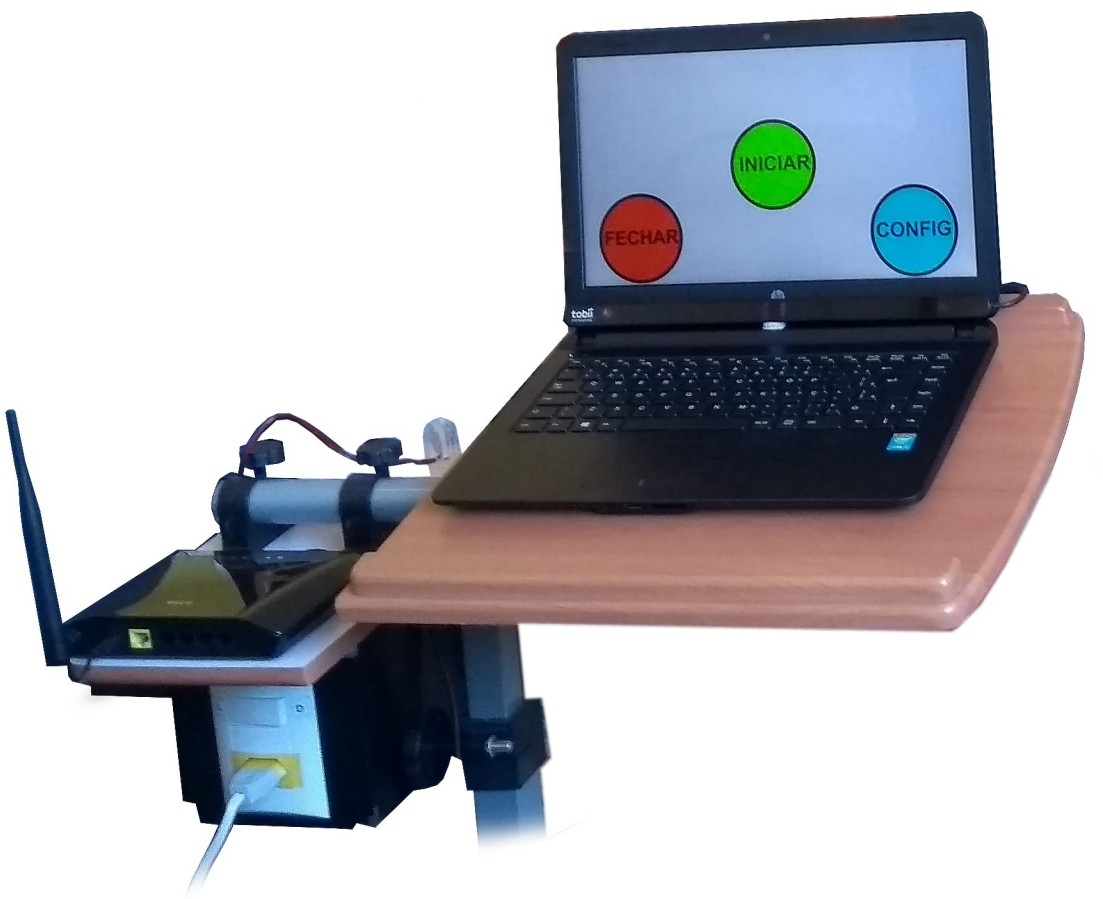

**Fig 3. Portable table, eye tracker, and notebook computer are installed.**

application in which it is possible to download the use data history, among other options. 6. Wi-Fi Router: to send the signal from the notebook computer to the gBox. 7. Portable table: used to position the equipment (Fig 3), facilitating its transport and use.

After clicking on START (a), the main menu (b) appears on the screen and the user can go to the icon associated with the device they wish to control (START), configure the system (CONFIG) (c), or return to the initial screen (CLOSE). To activate the devices (d), the mouse cursor must be positioned on desired icon for the time defined in the settings, then its background turns yellow (e), except for the TV icon, which has an individual submenu (f) to turn it on/off or control its channels or volume.

## Data collection instruments

The following instruments were used to collect information about the participants during HMI use:

Sociodemographic data forms: used to collect the participants' personal data, information on the diagnosis and history of the disease or injury, and experience with technology.

Functional Independence Measure (FIM$^{TM}$) [63, 64]: it assesses the degree of assistance needed for users to perform motor and cognitive ADL.

Canadian Occupational Performance Measure (COPM) [65]: it evaluates changes in the client's perception of their performance in activities and their satisfaction with them.

Psychosocial Impact of Assistive Devices Scale (PIADS) [66]: it assesses the effects of an AT device on the functional independence, well-being and QoL of users.

Quebec User Evaluation of Satisfaction with Assistive Technology (QUEST 2.0) [67, 68]: it measures satisfaction with the AT resource and the service delivered.

System Usability Scale (SUS) [69]: it evaluates the usability of the environment control system.

Semi-structured interviews: they were audio recorded, carried out to obtain information about the process of using the system (positive and negative points, and suggestions).

## Participants

Inclusion criteria: people aged ≥18 years with motor disabilities that compromised the normal interaction with equipment in the home environment, indicated by rehabilitation institutions or professionals; they should also have a caregiver of age and literate. Exclusion criteria: individuals with cognitive deficits (determined by their assisting professionals) that compromised understanding of the equipment functioning and use, as well as of the assessment instruments, and with visual deficits not corrected by glasses or contact lenses.

## Procedures

A visit to each of the participant's homes was scheduled for the initial assessment and installation of the system. After acceptance, each participant or caregiver signed a FICF, responded to the sociodemographic data form and the FIM[TM] and COPM measures. The FIM[TM] was applied through interviews [63] conducted by the main researcher, who is certified on the use of this instrument, and the COPM was applied directed to activities that require the use of equipment.

The system was installed in the residence room most used during the day as indicated by the participant or caregiver (Fig 4).

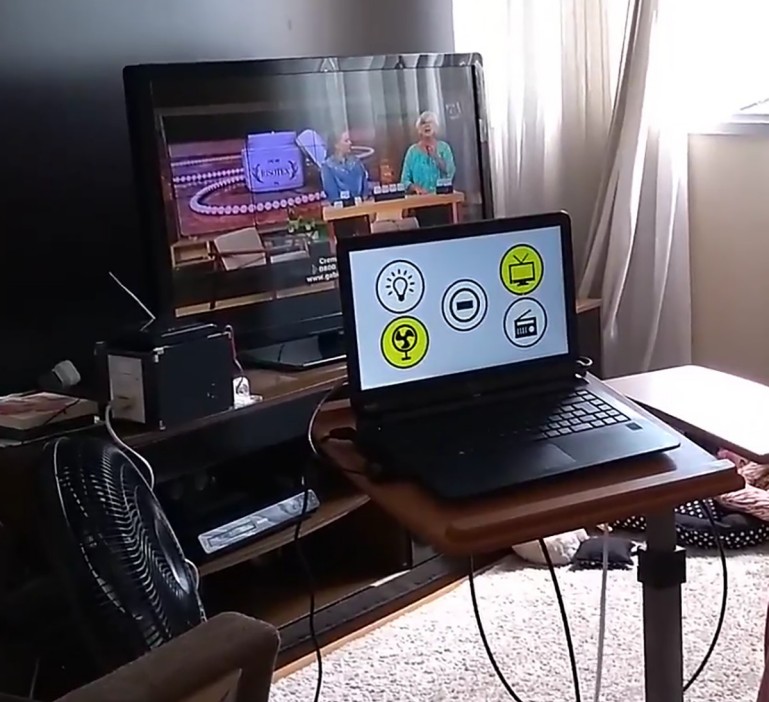

**Fig 4. System installed in the home of one of the study participants.**

Both the participant and the caregiver were trained to use the system and received a copy of the user manual containing explanations on the eye tracker calibration and equipment control. The caregiver's role was to turn on the notebook computer, position the portable table, and calibrate the eye tracker whenever necessary.

The system remained installed at each participants' homes for one week, as in the study conducted by Calvo et al. [70]. When necessary, extra visits to the participants' homes were made in order to make adjustments or assist with use.

At the end of that period, reassessments were carried out using the COPM, the other instruments (QUEST 2.0, PIADS, and SUS), and the semi-structured interview.

It is worth mentioning that a pilot study was previously carried out with a participant with motor disabilities aiming to verify the system functionality in a common home environment, and whether the methodology was adequate to the study objectives.

The pre- and post-test statistical analyses of the COPM instrument were performed using the Paired Sample $t$-Test considering a statistically significant difference of 5% ($p < 0.05$). This statistical test was chosen because of its robustness, considering the low sampling power obtained with small samples. Results of the other instruments are presented descriptively.

## Results

Six people with disabilities participated in this study. Of these, two individuals did not use the system during the period that the equipment remained in their homes, and their cases will be presented and discussed separately.

Table 1 shows the main information about the participants who used the system.

The mean age of participants was 49 years, ranging from 30 to 63 years. Participants PP1, PP5, and PP6 presented basic knowledge of technologies, more focused on the use of cell phones. Participant PP3, the youngest, had intermediate knowledge in using computers and cell phones before presenting the disease signs and symptoms.

Table 2 presents the results on functional independence collected through the FIM[TM].

As previously described, the FIM[TM] score considers the need for assistance in each activity. The participants' lower motor scores refer to difficulties in performing ADLs as well as in holding and manipulating objects used daily. Participant PP3 presented a lower cognitive score as a result of difficulty in communication.

In the COPM assessment, the participants were asked to indicate which activities were important in their everyday lives. Having 'control of the TV' was considered important by all participants, being able to 'turn the fan on/off' was deemed essential by two participants (PP5

**Table 1. Data of participants who used the system.**

| Participant | Gender | Type of caregiver | Health condition | Time elapsed since diagnosis |
|---|---|---|---|---|
| PP1 | F[a] | Informal caregiver[c] | ALS[e] + Psoriatic Arthritis | 5 months |
| PP3 | F | Informal + formal caregiver[d] | Autoimmune vasculitis | 8 years |
| PP5 | M[b] | Informal caregiver | SCI[f] (incomplete C7 level) | 29 years |
| PP6 | M | Informal caregivers | ALS | 1 year and 9 months |

[a] F–Female

[b] M–Male

[c] Informal caregiver–refers to a family member who cares the person

[d] Formal caregiver–refers to professionals who are paid to care

[e] ALS–Amyotrophic Lateral Sclerosis

[f] SCI–Spinal Cord Injury.

**Table 2. FIM[TM] results of participants who used the system.**

| Participant | Motor FIM[TM] | Cognitive FIM[TM] | Total FIM[TM] |
|---|---|---|---|
| PP1 | 47/91 | 35/35 | 82/126 |
| PP3 | 25/91 | 23/35 | 48/126 |
| PP5 | 44/91 | 35/35 | 79/126 |
| PP6 | 67/91 | 35/35 | 102/126 |

and PP6), and having 'control of the lights' was a significant demand for only one of the participants (PP1).

At reassessment, participants PP1, PP5 and PP6 responded to the instruments with the help of the researcher to make markings on paper, and the interview was answered without help. With regard to participant PP3, the interview was conducted with her mother and, for the other evaluations, the scales were designed in the notebook computer and the participant indicated the most appropriate response by moving the mouse cursor. The COPM was fully answered by the participant. The SUS and QUEST 2.0 were answered jointly with the participant's mother. Due to fatigue, her mother responded to the PIADS based on what she believed her daughter's responses would be.

Results of the COPM and the Paired Sample $t$-Test for all participants are shown in Table 3.

For the Paired Sample $t$-Test, only the events 'control of the TV' and 'COPM total score' were analyzed, as these events were common to all participants.

Statistically significant results were observed for performance and satisfaction regarding 'control of the TV' and for total satisfaction after using the system. Borderline results were obtained with respect to total performance.

**Table 3. COPM and Paired Sample $t$-Test results of participants who used the system.**

| Participant | Demands | Performance | | Satisfaction | | Change | |
|---|---|---|---|---|---|---|---|
| | | P1[a] | P2[b] | S1[c] | S2[d] | P2 Total -P1 Total | S2 Total -S1 Total |
| PP1 | Control of the TV | 7 | 8 | 5 | 8 | | |
| | Control of the lights | 6 | 7 | 5 | 8 | | |
| | Total score | 6.5 | 7.5 | 5 | 8 | 1 | 3 |
| PP3 | Control of the TV | 1 | 7 | 1 | 7 | | |
| | Total score | 1 | 7 | 1 | 7 | 6 | 6 |
| PP5 | Control of the TV | 1 | 10 | 5 | 10 | | |
| | Turn the fan on/off | 1 | 10 | 1 | 10 | | |
| | Total score | 1 | 10 | 3 | 10 | 9 | 7 |
| PP6 | Control of the TV | 3 | 9 | 3 | 10 | | |
| | Turn the fan on/off | 5 | 9 | 3 | 10 | | |
| | Total score | 4 | 9 | 3 | 10 | 5 | 7 |
| $p$-value[*] | | Performance | | Satisfaction | | | |
| Control of the TV | | 0.045 | | 0.009 | | | |
| COPM total score | | 0.050 | | 0.009 | | | |

[a] P1- initial performance

[b] P2- final performance

[c] S1- initial satisfaction

[d] S2- final satisfaction.

[*] Paired Sample $t$-Test ($p < 0.05$).

**Table 4. QUEST 2.0 scores.**

| Participant | Resource | Service delivery | Total |
|---|---|---|---|
| PP1 | 4.5 | 5.0 | 4.7 |
| PP3 | 4.4 | 5.0 | 4.6 |
| PP5 | 5.0 | 5.0 | 5.0 |
| PP6 | 5.0 | 5.0 | 5.0 |

Table 4 shows the results obtained with application of the QUEST 2.0 instrument. The results are very close or equal to 5.0 (the highest possible score), corresponding to high levels of satisfaction.

Table 5 presents the items that the participants considered most important about the SE control system. Each participant should indicate three items, and 'effectiveness' was pointed by three of the four participants as an important feature of the SE tested.

As for the PIADS instrument, Table 6 presents the score for each subscale and the final average of the participants, in which participants PP1, PP5, and PP6 were close to the maximum score (3.0).

Fig 5 illustrates the results of the SUS, whose average score was 85.6.

Through the interviews, all participants who used the SE system found it useful, mainly because it provided them with greater independence and exercise of autonomy in controlling the equipment, as it can be verified in some of their answers:

*"Ah, it is useful in all aspects, right? Turn on, turn off"* (PP1)

*"I think it was good. I think (PP3) was happy to get it, right? You saw her expression of joy, right? So, it was (useful). The part that I found most positive is giving autonomy, right? This is fundamental!"* (PP3's mother)

*"Its. . . accessibility to be able to move. (. . .) it was very useful. . . with the difficulty that I have (. . .). The facility for you to pick up and do things"* (PP5)

*"It brings independence! Not depending on anybody to 'turn up the volume!', 'Switch channels!', or 'turn on the television!', 'turn off the television!'"* (PP6)

As examples of difficulties or aspects that need to be improved in our system, the participants reported the delay to switch between distant TV channels; feeling tired or having a mild headache caused by the use of the eye tracker; dependence on a caregiver or family member to start the system and open the CIs; and the complicated process for calibrating the eye tracker.

The following suggestions were made: a numeric keyboard to type the desired channel; remove the need to use the notebook computer keyboard for some tasks, such as login to CI;

**Table 5. Important items regarding the SE system.**

| Item | Number of citations |
|---|---|
| Effectiveness | 3 |
| Adjustment | 2 |
| Simplicity of use | 2 |
| Professional services | 2 |
| Follow-up services | 1 |
| Comfort | 1 |
| Safety | 1 |

**Table 6. PIADS subscale scores.**

| Participant | Competence | Adaptability | Self-esteem | Average |
|:---:|:---:|:---:|:---:|:---:|
| PP1 | 2.6 | 3.0 | 3.0 | 2.9 |
| PP3 | 1.3 | 0.7 | 1.8 | 1.3 |
| PP5 | 2.5 | 3.0 | 2.6 | 2.7 |
| PP6 | 2.8 | 3.0 | 2.3 | 2.7 |

make the system simpler and more intuitive for people with little experience with computers; allow the system to also control all the home lighting of the residence and make and receive phone calls via a smartphone.

Regarding the user manual, participants PP1, PP5 and PP6 reported that they did not need to access it, because the explanation and training provided by the researchers were sufficient to use the SE system. Participant PP3's mother, on the other hand, reported that the manual did not clarify her doubts, requiring the presence of one of the researchers.

The system usage records, obtained through the Web application, enabled verification of the number of days that each participant effectively used the SE (Table 7).

Table 8 shows the data of the participants who did not use the system.

At the initial assessment using the COPM, both PP2 and PP4 reported that watching TV was a very important activity in their everyday lives, but that they were not satisfied with the way they performed this activity. Thus, the TV was the only device connected to the gBox for control.

Table 9 presents the results of the FIM$^{TM}$ with respect to functional independence.

According to the FIM$^{TM}$ data, both participants (PP2 and PP4) had need for maximum assistance to perform motor activities and presented total independence for cognitive activities.

At reassessment, these participants stated that they found the equipment useful, responding positively to all the assessment instruments. However, the system data records available at the Web application showed that they do not use the equipment at all.

Both participants present some similar characteristics that may have contributed to not using the equipment: they have difficulty dealing with the diagnosis or with their current health condition; caregivers not close or not engaged in this additional task; they report that

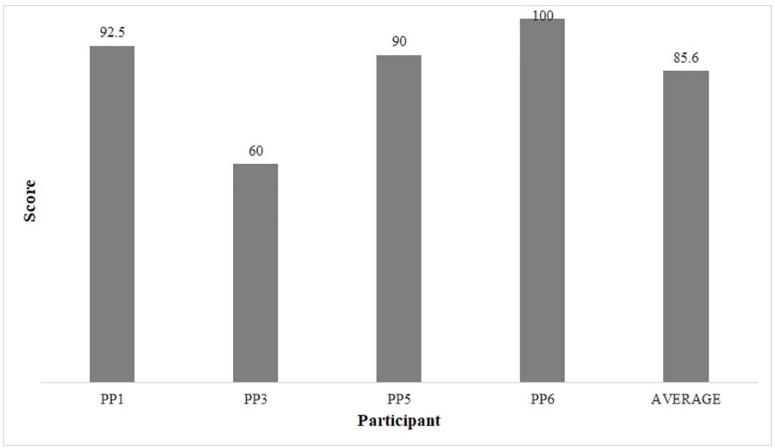

**Fig 5. SUS results of participants who used the system.**

**Table 7. Usage registration information obtained through the Web application.**

| Participant | Number of days of use |
|---|---|
| PP1 | 2 |
| PP3 | 5 |
| PP5 | 2 |
| PP6 | 4 |

the equipment does not allow total independence and that they have little knowledge of technology, limited to communication through the smartphone.

## Discussion

This study was conducted with six participants. Despite its small sample size, this research aimed to analyze participants using the system in their homes, for a prolonged time, and not occasionally in the laboratory, because its main objectives were to assess occupational performance, usability, and satisfaction with the developed AT system.

All the participants who used the SE system need considerable assistance from their families or formal caregivers to perform their ADL, which was evidenced by the FIM$^{TM}$. For them, 'control the TV' was the most important activity, according to the COPM.

The TV is an extremely popular appliance and an important resource of entertainment. According to Myburg et al. [71], TV control systems were among the most frequent environment control devices in the population studied (people with SCI).

Although the system presents more options to control electronic devices, the participants did not use all of them, according to the COPM. Several researchers have reported the significance of considering factors that are important for the people who will use an AT device [22, 24, 32] aiming at better adherence and results.

Two participants (PP1 and PP6) wear glasses. Duchowski [54] points out that the use of lenses (contact lenses or glasses) can interfere with the eye tracker ability to locate the corneal reflex, as they have reflective surfaces; however, the use of glasses did not interfere with the performance in using the system in the present study.

In the COPM, the participants self-evaluated their performance (P) in the activities they considered important, and their satisfaction (S) with performance before and after using the system. The higher the score, the better the performance or satisfaction.

The final assessments (P2 or S2) of all participants who used the system were higher than their initial assessments (P1 or S1). Except for the change in performance of participant PP1, all other evaluations showed changes greater than two points, which is considered by Law et al. [65] as a clinically important change.

Statistical analysis of the COPM showed positive results. For participants PP3, PP5 and PP6, the initial evaluation scores indicate that they were unable to perform the activities or presented great difficulty in performing them, also reflecting on their low satisfaction. At the final assessment, the results clearly showed that the participants had a new way of interacting with the environment more actively and, consequently, greater satisfaction with performance.

Among all the participants, participant PP1 was the only one who still has some manual skills, thus she can operate the TV remote control, although with some difficulty, and getting tired along the process. Therefore, she presented higher initial scores and smaller changes at reevaluation.

Regarding the QUEST 2.0, to assess the satisfaction with the resource, the participants should consider the entire set of hardware (gBox, notebook computer, eye tracker, router, and

**Table 8. Data of participants who did not use the system.**

| Participant | Gender | Type of caregiver | Health Condition | Time elapsed since diagnosis |
|---|---|---|---|---|
| PP2 | F[a] | Formal caregivers[c] | Multiple Sclerosis | 4 years |
| PP4 | M[b] | Informal[d] + formal caregivers | SCI[e] (C5 level) | 2 years |

[a] F–Female

[b] M–Male

[c] Informal caregiver–refers to a family member who cares the person

[d] Formal caregiver–refers to a professionals who are paid to care

[e] SCI–Spinal Cord Injury.

portable table) and software (CI). Participants PP1 and PP3 scored less than 5.0, referring to difficulties in calibrating the eye tracker, occasional visual discomfort, and difficulty in using the system (in the case of the participant with the greatest motor impairment).

To assess the satisfaction with the service provided, the participants considered installation of the equipment, explanations, training, troubleshooting, and necessary follow-up during the week of use. In this item, all the scores were the highest (5.0). Skilled professionals and services are items appointed by Lenker et al. [22] as an important point in the process of obtaining an AT resource, leading to better outcomes with use. In contrast, lack of continuous support can lead participants to lose interest in its use [17].

The QUEST 2.0 total average (between 4.6 and 5.0) obtained in this research shows that the participants were satisfied with the SE. This result corroborates the findings of two studies of the systematic review conducted by Brandt et al. [48] on environmental control systems and smart homes used by people with disabilities.

The aspect that the participants considered most important in the SE control system was 'effectiveness'. Demers et al. [72] defined this term as the "goal achievement with the AT device" (p.189), reinforcing that the system has met the needs of these people. Our findings corroborate those by Shone Stickel et al. [73], who also found effectiveness as the most important attribute of electronic AT devices for performance of ADL.

In the PIADS, respondents assessed how they were affected by the SE system. Participants PP1, PP5, and PP6 had the highest average values, indicating a maximum positive impact with the use of the SE. They assigned the highest values to the Adaptability subscale, indicating that with the use of the system they felt more willing to take risks and more motivated to participate socially [66]. Participant PP3, who has the most significant motor impairment, presented the lowest average among the participants. This instrument, as previously mentioned, was answered by her mother based on what she believed her daughter's assessment would be. Thus, it may not reliably represent the participant's assessment.

The developers of this instrument [66] claim that these three subscales (Competence, Adaptability, and Self-esteem) are sufficiently sensitive to assess the psychosocial impact of an AT device or resource on the user, which are included in the QoL concept. In addition, the longer the period of use, the greater the feeling of competence [74], being that the hypotheses for it are that the longer the usage time: 1) the more the users appreciate the effect; 2) reflects the user's real need for the device.

**Table 9. FIM™ results of the participants who did not use the system.**

| Participant | Motor FIM™ | Cognitive FIM™ | Total FIM™ |
|---|---|---|---|
| PP2 | 26/91 | 35/35 | 61/126 |
| PP4 | 13/91 | 35/35 | 48/126 |

Due to the short time of use of the SEs in this study (one week), it is not possible to state that there was a real change in the psychosocial aspects of the participants, but it indicates a tendency toward this change, in view of the results.

Regarding the SUS instrument, the result (85.6) indicates that the usability of our system was well evaluated. According to Bangor, Kortum and Miller [75], products evaluated in the range of 80 points are considered good, and products evaluated in the range of 90 points are considered exceptional.

The lowest evaluation refers, again, to participant PP3, whose answers pointed to some degree of complexity in the system and the need of a technical person or prior learning. Beyond the motor impairment of this participant, other factors may have interfered with the use of the eye tracker, such as her position in bed, small opening of the eye sometimes, fatigue with use, and difficulty of caregivers with the use of computers and programs.

Despite the lower ratings assigned by this participant, it seems that for all participants, on average, the assessment instruments showed positive results in relation to occupational performance, satisfaction with performance, satisfaction with the SE system, and usability of the system.

Many studies have evaluated improvements in these aspects after people with disabilities used environment control systems or electronic AT devices [17, 44, 73, 76, 77], whereas other studies have assessed ways of interacting with the environment through eye trackers [78, 79]. However, no studies with the same objectives and using the same methodology of the present research, that is, use of IROG technology for SE control, have been found for comparison.

As the results show better occupational performance, satisfaction with performance with the SE, and system usability, it can be concluded that the SE controlled by IROG evaluated in this research provided people with motor disabilities with more independent operation and control of the equipment.

All the reports of participants point positive aspects with the use of the SE system. These statements corroborate the researched literature [80–83], since independence, control and privacy are highly important aspects pointed by people with disabilities who used environment control systems or electronic aids to daily living (EADL).

Participant PP6's speech also points to an outcome present in the study by Verdonck, Chard and Nolan's [81]: the embarrassment that people with disabilities present regarding their recurring requests for help, followed by apologies, as they feel uncomfortable to interrupt their caregivers' routine. According to those authors, the use of EADL changes this dynamic, with fewer apologies, less discomfort, and reduced caregiver burden.

The user manual was an additional material left with the participants to assist with the use of the SE system. The literature highlights how important explanations and training are for understanding the use and for adherence to the AT resource. Myburg et al. [71] found that training was considered crucial for the total integration of the environment control system in the lives of people with spinal cord injury, as well as the involvement of the occupational therapist in the testing, prescription and configuration of the system.

Information obtained through the Web application showed that the system was not used every day. The justifications given by the participants included trips, medical or rehabilitation consultations, and other leisure activities, such as going to church or taking short tours.

However, some other hypotheses were raised, corroborating the literature: the system has limitations, requiring other person to activate part of the equipment [82, 84] or, when there is some voluntary movement, people prefer to behave as they are more accustomed [85]. Another possibility is that the TV was controlled by a person who was in the same room as the participant, using the standard TV remote control.

About the participants who did not use the system, Costa et al. [85] found some factors that can contribute to the understanding: lack of equipment functionality (for not providing the desired independence), difficulty in use, embarrassment in using the device, lack of support from family members, and lack of user motivation.

Regarding non-acceptance of diagnoses, studies have shown that this is an important factor to be considered when prescribing or selecting an AT device or resource [25, 86]; however, in the present study, this information only appeared at the reevaluation.

Wessels et al. [25] reported that there is a difference in the way the AT resource will be viewed between people who were born with a disability (for them, technology opens a new range of possibilities) and those who have acquired a disability (because, for them, technology will never replace the lost function).

Participants PP2 and PP4 are in the second group, since they acquired the disability as adults. A recurring line in the interviews is that they were very active and independent in the past, and now they are dependent for practically all activities. For them, the disability has also brought other types of losses, such as moving from their hometowns, changing their standard of living, ending relationships, or losing jobs. Such cases often result in periods of depression [25]. In this sense, for these people, it is hypothesized that they would only benefit from technology if their dependency could be completely reversed.

Another associated factor that may have contributed to non-use of the SE is that the AT device can highlight the disability [19, 23, 86]. Verza et al. [86] found that 30.3% of the reasons for abandoning or not using an AT device are due to the patient's non-acceptance. For those authors, although the AT device is seen as a possibility to increase functionality, it can be interpreted as a validation of the disability and loss of independence, resulting in decreased self-esteem.

It should be noted that, although the system registers activation of the equipment, this information was not passed on to the participants, so that the use of the system was based on their real needs and desires, and not on the fact that they felt obliged to use it.

It is worth restating that it is important that the professional involved perform a wide and in-depth assessment of the patient's real demands, expectations, and possibilities of the proposed AT device, as well as consider their participation in the choice. These points are important to ensure acceptance and continuity of use, since abandonment can represent a waste of resources (their own or from the government) [17, 24, 26, 27].

## Conclusions

Participants who used the IROG-controlled SE system showed better occupational performance and satisfaction with performance. In addition, the psychosocial impact was close to the maximum, satisfaction with the system was well evaluated and, for three participants, the usability was considered good.

The two participants who did not use the system presented characteristics such as non-acceptance of their diagnoses or current health conditions, and weak relationship with caregivers. Besides that, the fact that an AT device possibly reinforces disability may have corroborated these results.

The HAAT model was used as a basis for the study, considering "a person with motor disability (human) controlling the electronic equipment (activity) in its own house (context) using an SE system (assistive technology)".

Although the AT resource (SE system) was previously defined, in order to be evaluated, the most important activities for each participant were considered, and the most familiar place was used as research setting. The participants' motor and cognitive skills were considered, then the eye tracker using IROG technology was chosen as the less intrusive technique.

Considering that disability is indicated by the HAAT model and the ICF as inherent in social structures, and not in the person, the results found suggest that the SE system enabled reduction in the incapacity of the participants, who thus had greater participation in related activities.

This study provided a wide evaluation of equipment that aims to allow greater independence for people with severe motor disabilities, from the point of view of its operation and usability, as well as the benefits it provided to the people who used it. For professionals, this study highlights the importance of a good evaluation for the prescription and development of AT resources, avoiding abandonment or non-use.

A limitation to this study regards its small sample size (n = 4), whose statistical analysis does allow generalization of the results, which may hinder its reproducibility.

Future studies with larger samples and longer duration should be conducted, expanding the possibilities of controlled equipment and devices, in order to understand whether the benefits remain in long term.

## Patents

The gBox patent, together with the environment control system named "Remote micro-controlled device for charging residential loads via the Internet with emitter and receiver of commands via integrated infrared", was submitted to the Institute of Technological Innovation–INIT at UFES, Brazil, and evaluation is under process.

## Supporting information

**S1 Dataset.**
(XLS)

## Acknowledgments

The authors are grateful to all the professionals and participants involved in this study for their contribution.

## Author Contributions

**Conceptualization:** Mariana Midori Sime, Teodiano Freire Bastos-Filho.

**Data curation:** Mariana Midori Sime, Alexandre Luís Cardoso Bissoli, Daniel Lavino-Júnior.

**Formal analysis:** Mariana Midori Sime, Teodiano Freire Bastos-Filho.

**Funding acquisition:** Alexandre Luís Cardoso Bissoli, Teodiano Freire Bastos-Filho.

**Investigation:** Mariana Midori Sime, Daniel Lavino-Júnior.

**Methodology:** Mariana Midori Sime, Alexandre Luís Cardoso Bissoli.

**Project administration:** Mariana Midori Sime.

**Resources:** Mariana Midori Sime, Alexandre Luís Cardoso Bissoli, Daniel Lavino-Júnior, Teodiano Freire Bastos-Filho.

**Software:** Alexandre Luís Cardoso Bissoli, Daniel Lavino-Júnior.

**Supervision:** Teodiano Freire Bastos-Filho.

**Validation:** Mariana Midori Sime, Teodiano Freire Bastos-Filho.

**Visualization:** Mariana Midori Sime, Teodiano Freire Bastos-Filho.

**Writing – original draft:** Mariana Midori Sime.

**Writing – review & editing:** Mariana Midori Sime, Teodiano Freire Bastos-Filho.

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
