## [Decision Letter · Decision Letter 0]

3 Mar 2021

PONE-D-20-12562

Usability, occupational performance and satisfaction evaluation of a smart environment controlled by infrared oculography by people with severe motor disabilities

PLOS ONE

Dear Dr. Sime,

Thank you for submitting your manuscript to PLOS ONE. After careful consideration, we feel that it has merit but does not fully meet PLOS ONE’s publication criteria as it currently stands. Therefore, we invite you to submit a revised version of the manuscript that addresses the points raised during the review process.

For acceptance, it is crucial that you fully consider the comments and criticism of the reviewer.

We look forward to receiving your revised manuscript.

Kind regards,

Peter Schwenkreis

Academic Editor

PLOS ONE

Additional Editor Comments :

-the paper needs to be substantially shortened and condensed

-details and photos that might allow to identify the participants should be omitted

-"discussion" mus be separated from "results"

-Please explain in more detail why "data cannot be shared publicly because of Ethics Committee terms"

Journal Requirements:

2. We note that your paper includes detailed descriptions of individual patients/participants. As per the PLOS ONE policy (http://journals.plos.org/plosone/s/submission-guidelines#loc-human-subjects-research) on papers that include identifying, or potentially identifying, information, the individual(s) or parent(s)/guardian(s) must be informed of the terms of the PLOS open-access (CC-BY) license and provide specific permission for publication of these details under the terms of this license. Please download the Consent Form for Publication in a PLOS Journal (http://journals.plos.org/plosone/s/file?id=8ce6/plos-consent-form-english.pdf). The signed consent form should not be submitted with the manuscript, but should be securely filed in the individual's case notes. Please amend the methods section and ethics statement of the manuscript to explicitly state that the patient/participant has provided consent for publication: “The individual in this manuscript has given written informed consent (as outlined in PLOS consent form) to publish these case details”.

4. Thank you for stating the following in the Financial Disclosure section:

"AB and TBF received a award in Google Research Awards for Latin America, of Google Inc.

We note that you received funding from a commercial source: Google Inc.

Reviewers' comments:

Reviewer's Responses to Questions

**Comments to the Author**

1. Is the manuscript technically sound, and do the data support the conclusions?

Reviewer #1: Partly

2. Has the statistical analysis been performed appropriately and rigorously? 

Reviewer #1: Yes

3. Have the authors made all data underlying the findings in their manuscript fully available?

Reviewer #1: No

4. Is the manuscript presented in an intelligible fashion and written in standard English?

Reviewer #1: Yes

5. Review Comments to the Author

Reviewer #1: This is an interesting article, however it needs major modification prior to being eligible for publication. Each section needs to be condensed. While there is no word limit, PLOS ONE asks that papers be presented concisely - and this paper does not meet that criteria. For example, there is no need for a separate introduction and literature review - these should be combined and condensed. Similarly the description of the materials used could be reduced. The reference to "volunteers" should be changed to "participants" throughout the manuscript. I am also uncomfortable with the inclusion of the case study vignettes for each participant, I believe these have the potential to identify participants, especially in conjunction with their demographic data. There is no need for this level of detail. Also, the results and discussion sections need to be separated out. It is very unusual to see them presented together and does not meet the manuscript organisation guidelines specified for PLOS One (available here: https://journals.plos.org/plosone/s/submission-guidelines#loc-length). I am also not comfortable of the inclusion of photos of participants in their home environments. These are extremely identifying, even though the faces have been obscured.

6. PLOS authors have the option to publish the peer review history of their article (what does this mean?). If published, this will include your full peer review and any attached files.

Reviewer #1: No

---

## [Author Response · Author response to Decision Letter 0]

22 May 2021

Response to Reviewers

Title: Usability, occupational performance and satisfaction evaluation of a smart environment controlled by infrared oculography by people with severe motor disabilities

In response to the reported pending issues, the following information is required:

- Point 1: The paper needs to be substantially shortened and condensed.

Answer to point 1: As recommended by the reviewer, the introduction and review have been combined and condensed. The description of the materials and instruments has been reduced. The results and discussion were separated and reorganized.

- Point 2: Details and photos that might allow to identify the participants should be omitted.

Answer to point 2: The vignettes with the details and photos of the participants have been removed and only the sociodemographic information that contributed to the discussion was maintained.

- Point 3: About the funding received of the Google Inc.

Answer to point 3: Google Inc. only supported the research through Google’s Latin America Research Awards to two authors, Teodiano Freire Bastos-Filho and Alexandre Luís Cardoso Bissoli, and this does not alter our adherence to Plos ONE policies on sharing data and materials. This statement can also be found in the cover letter, in the Competing Interests Statement section, as recommended 

- Point 4: Please explain in more detail why "data cannot be shared publicly because of Ethics Committee terms".

Answer to point 4: Our data contained information that identified the participants. This information has been omitted and the data are being shared in the Supporting Information files.

- Point 5: Please amend the methods section and ethics statement of the manuscript to explicitly state that the patient/participant has provided consent for publication: “The individual in this manuscript has given written

informed consent (as outlined in PLOS consent form) to publish these case details”

Ansewer to point 5: About the use of personal data, participants or their guardians signed the Free and Informed Consent Form (FICF), which guarantees that all personal data are confidential and private, even after the publication of the research. The data protection has been guaranteed by identifying participants by codes, instead of their names or initials, and by not including information such as date of birth and address. Detailed information about the participants health condition had already been excluded from the manuscript after the original decision letter. The terms with the highlighted subsection in both the original version in Portuguese and the translated version in English were included in the list of documents for submission. The current situation in Brazil, due to the COVID-19 pandemic is very serious, and it has been very difficult to meet the participants to obtain their signature on the PLOS specific consent form. Since they are people with disabilities and comorbidities, they have been in quarantine, as a measure to prevent spread of the Sars-CoV-2 virus. The methods section and ethics statement of the manuscript were amended to explicitly state that the participants has provided consent for publication, through the FICF.

Sincerely,

Mariana Midori Sime (e-mail: mariana.sime@ufes.br) 

Vitória-ES (Brazil), May 01, 2021.

---

## [Decision Letter · Decision Letter 1]

14 Jun 2021

PONE-D-20-12562R1

Usability, occupational performance and satisfaction evaluation of a smart environment controlled by infrared oculography by people with severe motor disabilities

PLOS ONE

Dear Dr. Sime,

Thank you for submitting your manuscript to PLOS ONE. After careful consideration, we feel that it has merit but does not fully meet PLOS ONE’s publication criteria as it currently stands. Therefore, we invite you to submit a revised version of the manuscript that addresses the points raised during the review process.

In the revised version, you should further shorten the "conclusions" section. Besides, regarding the small sample size of the statistical analysis (n=4), it is necessary to add a paragrah where you discuss the potential limitations of the study with respect to generalizability and reproducibility.

We look forward to receiving your revised manuscript.

Kind regards,

Peter Schwenkreis

Academic Editor

PLOS ONE

Journal Requirements:

Reviewers' comments:

Reviewer's Responses to Questions

**Comments to the Author**

1. If the authors have adequately addressed your comments raised in a previous round of review and you feel that this manuscript is now acceptable for publication, you may indicate that here to bypass the “Comments to the Author” section, enter your conflict of interest statement in the “Confidential to Editor” section, and submit your "Accept" recommendation.

Reviewer #1: All comments have been addressed

2. Is the manuscript technically sound, and do the data support the conclusions?

Reviewer #1: Yes

3. Has the statistical analysis been performed appropriately and rigorously? 

Reviewer #1: Yes

4. Have the authors made all data underlying the findings in their manuscript fully available?

Reviewer #1: Yes

5. Is the manuscript presented in an intelligible fashion and written in standard English?

Reviewer #1: Yes

6. Review Comments to the Author

Reviewer #1: Thank you for addressing the comments provided to you through the first review. I believe this paper is worth publishing. I feel some minor amendments are now required. I have noted these in the tracked changes version of your manuscript. I also want to confirm how the FIM was administered, as it appears this was via self report from the way this section is written (the FIM must be administered by a credentialed assessor). I also feel the conclusion could be condensed further, and there could be further consideration of the limitations of this paper (case studies, difficult to generalise etc.)

7. PLOS authors have the option to publish the peer review history of their article (what does this mean?). If published, this will include your full peer review and any attached files.

Reviewer #1: No

---

## [Author Response · Author response to Decision Letter 1]

29 Jul 2021

In response to the reported pending issues, the following information is required:

- Point 1: The conclusion could be condensed further.

Answer to point 1: The Conclusions section has been shortened, as solicited. 

- Point 2: It is necessary to add a paragraph where the potential limitations of the study are discussed.

Answer to point 2: A paragraph describing the limitations of the study resulting from its small sample size has been included in the Conclusions section.

- Point 3: Explain how the FIMTM was administered.

Answer to point 3: Regarding the questioning of the reviewer about the application of the FIMTM, it has been included in the text that the instrument was applied by a certified researcher through interviews, as validated in Brazil. Validation of the FIMTM in Brazil was performed by Riberto et al. (2004), which is reference no. 63 in the paper.

---

## [Editor Report · Decision Letter 2]

2 Aug 2021

Usability, occupational performance and satisfaction evaluation of a smart environment controlled by infrared oculography by people with severe motor disabilities

PONE-D-20-12562R2

Dear Dr. Sime,

We’re pleased to inform you that your manuscript has been judged scientifically suitable for publication and will be formally accepted for publication once it meets all outstanding technical requirements.

Kind regards,

Peter Schwenkreis

Academic Editor

PLOS ONE
---

## [Editor Report · Acceptance letter]

5 Aug 2021

PONE-D-20-12562R2 

Usability, occupational performance and satisfaction evaluation of a smart environment controlled by infrared oculography by people with severe motor disabilities 

Dear Dr. Sime:

I'm pleased to inform you that your manuscript has been deemed suitable for publication in PLOS ONE. Congratulations! Your manuscript is now with our production department. 

Kind regards, 

on behalf of

Dr. Peter Schwenkreis 

Academic Editor

PLOS ONE